Determining the optimal areas of effort in terms of force and force-velocity based on the functional state of the neuromuscular system in the training of elite female judokas

Manolachi Veaceslav 1 2
Potop Vladimir 2 3
Manolachi Victor usefs.pps@mail.ru 1 2
Delipovici Irina 2
Liuşnea Cristian Ştefan 1
1 Dunarea de jos University of Galati , Galati , Romania
2 State University of Physical Education and Sport , Chisinau , Republic of Moldova
3 Depatment of Physical Education and Sport, University of Pitesti , Pitesti , Romania
Georgian Badicu
Electronic publication date: 2022 May 20
Publication date: 2022
Volume: 10
Electronic Location ID: e13468
Received 2022 Feb 28; Accepted 2022 Apr 29
Copyright: ©2022 Manolachi et al.
Copyright year: 2022
Copyright holder: Manolachi et al.
License: This is an open access article distributed under the terms of the Creative Commons Attribution License, which permits unrestricted use, distribution, reproduction and adaptation in any medium and for any purpose provided that it is properly attributed. For attribution, the original author(s), title, publication source (PeerJ) and either DOI or URL of the article must be cited.
License URL: https://creativecommons.org/licenses/by/4.0/

Keywords: Elite athletes, System of training corrections, Functional state, Muscle activity, Diagnosis, Specific physical training, Statistical analysis

Funding: The authors received no funding for this work.

==============================
Background

The evaluation of the functional state of the neuromuscular system (NMS) in elite female judokas according to the muscular activity indices is influenced by the force (F) and force-velocity (F-V) efforts. The implementation of the individual correction plan in the elite female judokas’ process of instruction and training must be based on the accurate determination of the force and force-velocity effort areas throughout the training and competitive periods.

Methods

The research involved 44 elite female judokas, divided in to two experimental groups (A, C) and two control groups (B, D). To diagnose the NMS, 21 elite female judokas were evaluated, belonging to different classification categories, divided into three groups: group I–category I (Cat. I), group II–candidates for Masters of Sports (CMS) and group III—Masters of Sports (MS).The evaluation of NMS was performed at the end of each two-week cycle, using 3 tests: Tmax—time to reach the maximum muscle contraction, (msec); Fy –examination of elasticity indices in different muscles, (Hz); J–explosive muscular strength, (kg/s). Measurements were made for six muscle groups and 9 indices of fitness tests: 5 force tests (F) and 4 force-velocity tests (F-V). The research was carried out during 24 cycles, each one lasting two weeks: 12 training cycles and 12 competitive ones. Each cycle lasted two weeks. During the research, the model plan of training for F and F-V was used, determining the optimal areas of F and F-V training efforts in the preparation and competitive periods. In the experimental groups (A and C), according to the data of judokas’ NMS functional state evaluation, the individual correction of the F and F-V efforts was performed every two weeks. As for the control groups (B, D), traditional methods of training were used.

Results

The NMS evaluation of the female judokas was made every two weeks and the level of correlation of Tmax, Fy and J indices was determined. The value of the Fy index at F effort is 32% in group I, 30% - in group II, 27% - in group III, 28–30% at the effort of F-V. The total number of corrections in group A: 79 negative corrections and 59 positive corrections, while in group C: 65 negative corrections and 89 positive ones. Within the F-V effort, the number of effort corrections in group A was: 68 negative corrections and 92 positive; in group C - 81 positive and 78 negative corrections. The female judokas’ results in the final stage of EG–A were significantly improved in F indices by 52.15%, F-V by 6.22% and 6.18%. In the EG–C, the F increased considerably by 7.52%, F-V by 5.67% and 7.20%. These results characterize the level of physical training.

Conclusion

The functional state of the NMS in elite judokas, determined according to the temporal indices of reaching the maximal muscular contraction, the muscle explosive strength and the muscles elasticity, is subjected to dynamics under the influence of force effort and force-velocity effort.

Introduction

Sports training is a long-term adaptation to sports activities. For creating the most comfortable and efficient process of training, it is generally necessary to take into account the dynamics of athletes’ functional capacities in different periods of preparation. The current approaches to training process planning are very valuable in this regard. Concurrently, the mechanisms of adaptation, which are the basis for obtaining and developing the optimal effect of training remain insufficiently studied in many branches of sports (Verhoshansky, 2013; Platonov, 2015; Platonov, 2017; Manolachi, 2018).

In performance judo, the use of biomechanical measurements, fitness tests and special fitness tests (specific to judo), the investigation of the blood lactate concentration, the evaluation of heart rate and the perceived effort in a competition can provide an optimal control of training (Bonitch et al., 2005; Boguszewska, Boguszewski & Buśko, 2010; Branco et al., 2013). The evaluation of sports performance in judo, the Special Judo Fitness Test (SJFT), was considered an appropriate and comprehensive testing tool at all levels of judo practitioners, as well as for the athletes who do sports similar to judo (Drid, Trivić & Tabakov, 2012; Šimenko & Karpljuk, 2016; Casals et al., 2017; Kons et al., 2021). To establish the normative data for SJFT in female athletes, a systematic revision and meta-analysis was conducted; it provides data that can be used in the creation of the training programs for female judokas, as well as in the post-training evaluation and competitive training (Sterkowicz-Przybycien & Fukuda, 2014). The relationship between aerobic power, anaerobic power and the SJFT was studied in the elite male judokas (e.g., Hesari et al., 2014).

One of the main problems of sports training is the differentiation of physical activity depending on the adaptive skills of the body. Opportunities to improve and streamline sports preparation are provided by introducing science based management methods into the training process, and using the information technologies for the evaluation of the functional state of athletes’ neuromuscular system (NMS) (Bashkin & Kharlamov, 2015; Kharlanov, 2015; Fort-Vanmeerhaeghe et al., 2016; Kharlanov, 2021). The changes in the maximum muscle torque and maximum power of the lower limbs in male judokas during the pre-competitive training period revealed topographic modifications of the muscle groups studied (Buśko & Nowak, 2008). The strength of thigh muscles in terms of flexion/extension ratio must be evaluated in judo athletes in order to find out if there is an imbalance between the two muscle groups (e.g., Drid et al., 2010).

Neuromuscular fatigue is the result of neuromuscular activity. Quantitative assessments of the peripheral neuromuscular fatigue are made using electromyograms (Korjagina & Roguleva, 2018). In order to ensure the timely prophylaxis of athletes’ over-fatigue and to optimize the training process, an important factor is the development of methods for rapid and reliable diagnosis of the functional state of athlete’s body in training and competitions (Detanico et al., 2015; Franchini et al., 2019).

In general, the multi-annual training system for judokas should not notably differ from the same training system for the representatives of other combat sports. Achieving the best performance requires high levels of muscle power, accompanied by a moderate use of the aerobic metabolic pathway (Serrano et al., 2001; Sbriccoli et al., 2007).

Over the last few decades, women’s sports have progressed quite intensely. This is determined both by the social status of women and by their continuous tendency to assert themselves and be equal to men. The presence in the professional sports of women in the pre-involution period reveals and updates, for the physiology of age and the sports practice, the functional capacity of female body’s cardiovascular system to adapt to an intense physical activity at such a mature age. It is necessary to take into consideration the osteogenic effects of judo in the female practitioners during the premenopausal and postmenopausal period (Pogodina & Aleksanyants, 2017; Akhmetov et al., 2018; Ciaccioni et al., 2019).

The corrections must be applied in the instruction and training process of elite female judokas following an accurate determination of the effort areas related to force and force-velocity (e.g., Manolachi, 2015). It is also necessary to make time-movement and technical-tactical analysis (Miarka et al., 2014; Dudėnienė et al., 2017), to study the impact of the combat area size (e.g., Ouergui et al., 2021) and to monitor individually the hemodynamic parameters in response to loads of different capacities (e.g., Safarova, Pulatova & Sultanov, 2017). A relevant matter is the individualized improvement of judokas’ force-velocity training, based on the adaptive skills of the body (e.g., Fedorova, 2021). The training process in judo must take into account the recommendations on the physical preparation planning and organization and also the necessary changes in the training programs (Shepetyuk, Dzhambyrbaev & Ibrayev, 2017; Dadabayev, 2021). These aspects must be carefully addressed as women are currently full participants in various competitions of the highest level and their achievements in a number of sports are almost equal to men’s achievements. In the last decade many publications presented ideas on the directed improvement of the instruction and training process in order to increase the performances of women in different sports (Houvenaeghel et al., 2005; Franchini & Takito, 2014).

Women’s training should be planned differently than men’s training, even in the same sport, given the characteristics of the female body. The theory and practice of judo are evolving quite intensely, but most scientific investigations essentially reflect the particularities of sports training only in the male judokas (e.g., Kons et al., 2018). At the same time, the very dynamic practical development of women’s judo exceeds by far the quantitative and qualitative research in this field. There are, however, studies regarding the following matters: possibilities to concentrate the power training of elite female judokas throughout an annual training cycle; how to use the 2-D and 3-D video analysis for examining the muscle activity during a backheel (one of the basic techniques in judo); the concept of long-term force training in women’s judo (Eliphanov, 2011; Elipkhanov, Nemtsev & Kozlov, 2012; Elipkhanov, 2013).

From our point of view, the specialized scientific literature provides an insufficient number of works dealing with the preparation of the force and force-velocity effort in the elite female judokas. Unfortunately, to date, the degree of judo influence on the female body, the functional state level of the neuromuscular system (NMS) in the female judokas and the determination of the optimal effort areas in training and competitive periods have not been established with sufficient precision.

Purpose of the research. Determining the optimal areas of effort in terms of force and force-velocity according to the changes in the functional state of the neuromuscular system in the process of elite female judokas’ instruction and training.

Research hypotheses. The correlative analysis of the basic indices concerning the NMS functional state in the elite female judokas will establish the weight between the evaluated indices of the main muscle groups in each classification category investigated.

We also consider that the implementation of the individual correction plan in the instruction and training process will have to be based on the exact determination of the areas of force efforts and force-velocity efforts in the training and competitive periods.

Materials & Methods

Participants

The research involved 44 elite female judokas, belonging to different weight categories (48 kg, 52 kg, 57 kg, 63 kg, 70 kg, 78 kg and +78 kg), with an average and standard deviation of the age of 22.64 ± 1.53 years. The judokas were divided into four groups: two experimental groups (EG) and two control groups (CG). Depending on the level of sports classification, the judokas were distributed as follows: EG (A), n = 11 and CG (B), n = 11 with athletes of category I (Cat. I) and candidates for Masters of Sport (CMS) while the EG (C), n = 11 and CG (D), n = 11 included athletes in the category of Masters of Sport (MS). For the diagnosis of the neuromuscular system (NMS) functional state, 21 elite female judokas were selected, divided into three groups as follows: group I, n = 7 – judokas (Cat. I), group II, n = 7 – (CMS), group III-a, n = 7 – (MS), aged between 20–25 years. According to the Unique Sports Classification of the Republic of Moldova for the years 2018–2021, in order to be included in Cat. I, the judoka must take up to the 6th–7th place in the Championships of the Republic of Moldova (CRM); for the CMS position—she must win the 1st up to 5th place in CRM standings; for MS category, the 6th to 9th place in the standings of the European Seniors Championships.

The experimental study was approved by the Ethics Commission of the State University of Physical Education and Sports of the Republic of Moldova, in accordance with the Ethical Standards of the Declaration of Helsinki (ec_usefs4-11-2021). The participants gave their written consent for the study in conformity with the recommendations of the Biomedical Research Ethics Committees (World Health Organization, 2000).

The research was carried out in a laboratory provided with certified and accredited equipment.

Methods

The functional state of NMS in the female judokas was checked by means of three tests: Tmax–the time to reach the maximum muscle contraction, evaluated by the method of selective electromyography (msec); Fy–examination of the elasticity indices of different muscles, which was measured through the method of seismo myotonometry (Hz); J - explosive muscular strength, which was evaluated with the tensometric platform and the modular system of tensometric equipment (kg/s).

The instrumental methods used to obtain specific information on the functional state of the female judokas’ neuromuscular system (NMS) included the following research methods (Bashkin & Kharlamov, 2015; Kharlanov, 2015; Kharlanov, 2021; Kim & Podlesnykh, 2016):

(a) Method of selective electromyography (EMG): it was applied to determine the time to reach maximum muscle contraction. EMG was performed using flat electrodes attached to the following muscles: gastrocnemius; rectus femoris; biceps femoris; biceps brachii and triceps brachii. The test was made immediately after the warm-up. Prior to starting each study, the maximum possible level of the bioelectric activism of female athletes’ muscles was assessed. To this end, the maximal level of muscles activism was maintained (3 s) under the conditions of their isometric contractions and the retroactive visual control on the oscilloscope screen (KINE®, KINE ltd., Hafnarfjördur, Iceland). In order to determine the necessary time for reaching the maximum contraction, the athletes were requested to contract the studied muscle intensely and quickly (Bashkin & Kharlamov, 2015).

(b) Method of seismomyotonography: it was used to assess the functional state of athletes’ peripheral NMS according to the elasticity index of the following muscles: gastrocnemius, rectus femoris, biceps femoris, latissimus dorsi, biceps brachii and triceps brachii. The frequency of the muscle mechanical vibrations after the adjusted striking of the muscle was used as an index of elasticity. The frequency of vibrations was measured by means of the SKG seismic transmitter (Seismocardiograph 2000). The transmitter was attached to the analyzed muscle. The signal from the transmitter was sent to the computer, where the information was subsequently processed using a special program. The testing was carried out before the training sessions throughout the entire period of research (Muzykantova, 1984; Bashkin & Kharlanov, 2015; Kharlanov, 2021).

(c) Method of tensometry. The explosive power of the leg’s extensor muscles (such as tibialis anterior, extensor digitorum longus; extensor hallucis longus) and feet flexor muscles (such as gastrocnemius; soleus; plantaris; flexor digitorum longus) was established by means of the tensometric platform. The athlete, standing on the platform, jumped vertically as high and fast as possible from semi-squat position. The transmitters needed to enter the information into the computer system were located on the lower part of the tensometric platform. The explosive power of the biceps brachii, triceps brachii and back muscles was determined with the help of the tensodynamic system, which was connected to the tensometric platform. During the test, the athlete—with straightened legs, back slightly bent, both hands upwards (brief explosive strength of the back muscles)—had to rapidly perform a cable push-pull (triceps brachii or biceps brachii). The signal sent by the tensometric transmitters was directed to the computer system, for further processing (e.g., Steblecov, 1999).

(d) Pedagogical-sports testing. Tested indices: force (F) and force-velocity (F-V) (e.g., Manolachi, 2015).

(d.1) The tensometric device was used in the research for assessing the preparation level of the muscle force. This device determined the force of the back lumbar muscles (F1): the athlete-subject of the research stretched her legs and arms, with her torso inclined at an angle of 130 degrees when starting the pulls. For measuring the force of the thigh extensor muscles (F2), the athlete bent her legs at an angle of 160 degrees at the beginning of the pulls, while her arms and torso were straight. The force of the feet extensor muscles (gastrocnemius) was found out by standing on tiptoes, with legs, torso and arms straightened (F3). To determine the force of the biceps brachii, the athlete did pulldowns with the arms bent at 110 degrees and straight legs and torso (F4). The force of the triceps brachii is determined while the arms are bent with elbow joints at 110 degrees; the tensometric force transducers were placed at shoulders height of the studied subjects (F5). The explosive strength is determined when the test is performed on the tensometric platform, with 160 degrees bent legs at the knee joints (F-V1).

(d.2) Five Morote Seoi-nage (two handed back-carry throws) (F-V2). In order to standardize the task of the test, all participants performed the same type of throw (two-handed back-carry throws). The testing procedure was established in accordance with the recommendations of the specialists Yu.P. Zamjatin, B.I. Tarakanov, 1985, who used this test in wrestling. The testing procedure is described below (Zamjatin & Tarakanov, 1985). The judoka, who stood on the tatami in a face-to-face fighting stance with her opponent (of approximately the same body weight), made the necessary grip to perform the first throw. At the signal of the whistle, when the stopwatch started, the athlete made five Morote Seoi-nage as quickly as possible. After each throw, her partner had to rapidly return to the fighting position. The stopwatch was stopped at the end of the fifth throw and the displayed time was the result of the test. During the test, a group of experts visually evaluated the technique; the observance of the throw basic structure was used as a criterion of correctness, on condition that the grip was properly made and the throw included the flight phase of the opponent (e.g., Manolachi, Manolachi & Mruţ, 2019).

(d.3) Five Tai Otoshi (body drops) (F-V3). The structure of the technique performed in this test differs essentially from the one described in the previous test. At the same time, the procedure for testing and determining its results is exactly the same as the one presented above.

(d.4) Rope climbing (F-V4). The tested judoka stood on the tatami and grabbed the rope with both hands. At the whistle signal and the simultaneous start of the stopwatch, the athlete began to climb the rope without the help of her legs. A marking was made on the rope, 4 m above the tatami; the judoka was supposed to touch by hand this sign as fast as possible. When the marking was touched, the stopwatch was stopped and the result of the test was estimated with 0.1 s margin of error.

Experimental design

The research was conducted over a period of 24 two-week cycles (12 training cycles and 12 competitive ones), in two stages (E):

- E1 (training): the basic parameters of the NMS functional state in all muscle groups were evaluated depending on the size of the force effort and force-velocity effort in the judokas from groups I, II and III. The judokas belonging to the experimental and control groups (A, B, C, D) were submitted to the initial pedagogical-sports testing of the force (F) and force-velocity (F-V) indices.

- E2 (competitive): the basic parameters of the functional state of NMS in all muscle groups were evaluated according to the size of force effort and force-velocity effort in the judokas belonging to groups I, II and III. The judokas from the experimental and control groups (A, B, C, D) had the final pedagogical-sports testing of the force (F) and force-velocity (F-V) indices.

The model training plan for F and F-V was applied throughout the research. The following exercises were used for the efforts related to force: (1) standard pushups “all the way up and all the way down”; (2) behind the neck press (straightened lower limbs, arched back) on the “Hercules” exercise machine; (3) barbell behind neck squats, maximal weight. These exercises composed a single set. The following exercises were performed for the force-velocity efforts: (1) standard pushups in 10 s; (2) behind the neck press (using 50% of the own body weight) in 10 s; (3) barbell behind neck jumps from semi-squat position, in 10 s. These exercises too formed a single set. The analysis of the judokas’ NMS was made every two weeks of training. The minimum number of sets in the exercises related to force was 10, and the maximum number was 35, performed throughout two weeks. The minimum number of sets in the exercises related to force–velocity, along two weeks, was 15, while the maximum number was 45. The force effort was reduced by 10–50% (group A) and by 10–45% (group C); the force-velocity effort was reduced by 10–45% within the correction and increased by 10–50%.

Based on these model-characteristics and the training plan, the optimal areas of force and force-velocity training efforts were established in the training and competitive periods during the training annual cycle. In the experimental groups (A and C), in accordance with the data of judokas’ NMS functional state evaluation, every two weeks, the individual correction of the effort and effort-velocity efforts was performed (Table 1).

Table 1 Number of corrections of the force effort and force-velocity effort applied in the training of the female judokas of the experimental group A (categories I and CMS, n = 11) and group C (MS category, n = 11).

No. cycle	Effort	Group A	Group C	
		+	-	+	-	
I	F	–	5	–	5	
F-V	4	2	1	4	
II	F	3	11	7	11	
F-V	12	5	6	7	
III	F	4	4	12	1	
F-V	10	11	6	5	
IV	F	4	5	2	7	
F-V	10	8	4	10	
V	F	8	4	5	3	
F-V	5	5	6	6	
VI	F	9	7	11	8	
F-V	7	4	5	7	
VII	F	4	4	5	5	
F-V	4	4	8	5	
VIII	F	5	4	8	2	
F-V	7	6	7	3	
IX	F	5	8	7	6	
F-V	9	8	9	8	
X	F	5	13	11	4	
F-V	8	4	12	10	
XI	F	6	7	10	9	
F-V	12	4	11	9	
XII	F	6	7	11	4	
F-V	5	8	6	4	
Total	F	59	79	89	65	
F-V	92	68	81	78	
Notes.

F, force; F-V, force-velocity; positive (+) corrections; negative (-) corrections.

Totals for each row are shown in bold.

The plus sign (+) shows that the positive correction of the force effort and force-velocity effort was performed for each athlete and predetermines the increase of the training effort. The negative sign (-) expresses the achievement of the negative correction of the force efforts and force-velocity effort for each judoka; it means the diminution of the effort. The number of positive and negative corrections for each athlete within a monthly cycle change from 0 to 3 (in conformity with the number of functional parameters—Tmax, Fy and J). Instead, the control groups of female judokas (B, D) worked with traditional training methods.

Statistical analysis

All statistical analysis of the study results was performed using the SPSS (IBM Co, Armonk, NY, USA), version 23. Descriptive statistics methods were used to calculate mean, standard deviation (SD) and Cohen’s d effect size. The Parametric Test Pearson’s correlation coefficient was used to examine relations between variables and Independent-Samples T-test. Statistical significance was set at p < 0.05.

Results

The evaluation of the female judokas’ NMS functional state was performed every two weeks. It allowed to determine the level of correlation between Tmax, Fy and J indices of the main muscle groups and to balance the efforts of force and force-velocity.

Figure 1 shows the percentage value of the basic parameters of NMS functional state in all muscle groups depending on the force effort size. The data shown highlight the fact that when the level of athletes’ sports mastery increases, the value of Tmax index increases from 23% in group I to 41% in group III; inversely, the value of Fy index decreases from 40% to 32% in groups I and III, while the value of J index decreases from 32% to 27% in groups I and III. Therefore, one can notice significant differences in the percentage ratio of the basic parameters value at the same time with the increase of the female judokas’ classification level from category I to the Master of Sports title (MS).

Figure 1 (A–C) Percentage ratio of the value of the neuromuscular system indices of the female judokas with different levels of qualification, depending on the force effort.

Tmax–time to reach the maximum muscular contraction; Fy–indices of elasticity of different muscles; J–explosive strength of muscles.

The correlative analysis enabled also to determine the comparative percentage ratio of the parameters of NMS functional state in the female judokas, depending on the force and force-velocity training efforts.

Figure 2 presents the percentage value of the basic indices of NMS functional state of the sportswomen, according to the force-velocity effort. Thus, it was noticed that once the level of judokas’ sports mastery increases, the value of Tmax index goes up from 31% in group I to 39% in group III; the value of the Fy index decreases from 41% in group I to 31% in group III; the value of J index increases insignificantly from 28% in group I to 30% in group III.

Figure 2 (A–C) Percentage ratio of the value of the neuromuscular system indices of the female judokas with different levels of qualification, depending on the force-velocity effort.

Tmax–time to reach the maximum muscular contraction; Fy–indices of elasticity of different muscles; J–explosive strength of muscles.

Figure 3A shows the optimal areas of force training effort (Hc) depending on the value of the Tmax index of the rectus femoris muscle (curves are denoted by figure “1”) and of the gastrocnemius muscle (curves are denoted by figure “2”). The figure illustrates the directly proportional exponential dependencies of Tmax indices and Hc. The range of each optimal area is determined by the dispersion value of the data obtained in the group and constitutes approximately 25 msec. Within the optimal area, the force effort made during a two-week training cycle is shown in relation to the Tmax index of the respective muscle. The area of negative correction of the force training effort (in Fig. 3A it is denoted by the sign “-”) is situated above the optimal area. The area of positive correction (denoted by the sign “+”) is situated below the optimal area. The negative correction represents the decrease of the force training effort to the size of the value deviations from the optimal effort area, multiplied by the corresponding value coefficient. During the two-week training cycles, the force effort ranged from 10 sets of exercises to 35 sets with a corresponding increase of Tmax index of the rectus femoris muscle from the minimal value—110 msec up to the maximal value—250 msec; as for the gastrocnemius muscle—from 60 to 200 msec. Figure highlights the optimal areas of force-velocity training effort depending on the Tmax index of the rectus femoris (1) and the gastrocnemius muscle (2). The force-velocity effort during the two-week training cycles included 15 to 45 sets of exercises. The Tmax index of the rectus femoris muscle evolved from 100 msec to 230 msec, respectively, starting with group III and ending with group I; the Tmax index of the gastrocnemius muscle changed from 60 msec to 170 msec in those groups. The range of each optimal zone is about 25 msec. The positive correction area is distributed above the optimal areas and the negative correction area is located below the optimal areas.

Figure 3 Optimal areas of force effort and force-velocity effort based on the T_max indices of the tested muscles.

(A) 1–optimal effort area of the rectus femoris muscle; 2–optimal effort area of the gastrocnemius muscle; (B) 1–optimal effort area of the biceps femoris muscle; 2 - optimal effort area of the biceps and triceps brachii muscles; (C) optimal effort area of the latissimus dorsi; groups: I, II, III.

Figure 3B shows the optimal areas of force training effort, depending on the Tmax index of the biceps femoris, biceps and triceps brachii muscles (the curves are numbered accordingly, with “1” and “2”). The data presented in the figure show the directly proportional exponential dependencies of the Hc parameters on the Tmax index. The Tmax index of the biceps femoris increased from 105 to 240 msec, which corresponds to the force effort in 10 to 35 sets of exercises. The Tmax indices of the biceps and triceps brachii were almost identical, that is why their optimal areas coincide in Figure. The Tmax index value of the biceps and triceps ranged from 40 msec to 150 msec, which also corresponds to the minimal and maximal number of exercise sets. Figure presents the optimal areas of force-velocity training effort depending on the Tmax of the biceps, triceps and biceps femoris. The Tmax index of the biceps femoris oscillated from 110 up to 230 msec in the groups III–I. The Tmax indices of the biceps and triceps coincided, so they were presented as identical optimal areas. They ranged from 50 msec to 150 msec, which corresponds to the force-velocity effort in 15 to 45 sets of exercises.

Figure 3C shows the optimal areas of the force effort determined by the Tmax index of the latissimus dorsi. As shown in Figure, the Tmax index of the latissimus dorsi changed in group I from 160 msec to 260 msec, in group II from 140 msec to 240 msec and in group III—from 120 msec to 200 msec. The range of each optimal area is about 30 msec. Figure highlights the optimal areas of force-velocity training effort related to Tmax index of the latissimus dorsi. The Tmax index changed from 160 msec to 260 msec in group I, from 140 msec to 240 msec in group II and from 120 msec to 220 msec in group III. The range of each optimal area was approximately 30 msec. The area of negative correction is situated above the optimal areas while the area of positive correction is located below the optimal areas.

Figure 4 presents the optimal areas of the force training effort according to the Fy index of the tested muscles (biceps brachii, triceps brachii, gastrocnemius, rectus femoris, biceps femoris). The data summarized in Fig. 4 reveal the directly proportional exponential dependencies of the Hc and Fy indices. The range of each optimal area is determined by the value of data dispersion in the group and represents approximately 1.5 Hz. The Fy index ranged between 28 Hz and 35 Hz in group I, between 31 and 37 Hz in group II and between 32 and 39 Hz in group III. The area of negative correction is above the optimal areas and the area of positive correction is placed below the optimal areas. The force training effort progressed from the minimal value (10 sets of exercises) to the maximal one (35 sets of exercises). Figure shows the optimal areas of the force-velocity training effort depending on the Fy index of the tested muscles (biceps brachii, triceps brachii, gastrocnemius, rectus femoris, biceps femoris). The optimal areas are exponential curves, each curve having a range of approximately 1.5 Hz. The Fy index evolved from 28 to 35 Hz in group I, from 30 to 37 Hz in group II and from 32 to 39 Hz in group III. If the Fy indices values of the tested muscles exceed the 40 Hz threshold, then the risk of traumas in the respective muscles increases significantly, especially at the force-velocity efforts.

Figure 4 Optimal areas of force effort training and force-velocity depending on the Fy indices of the tested muscles.

OHc, Optimal areas of force effort training; OHcc, Optimal areas of force-velocity effort training.

Figure 5A highlights the optimal areas of the force training effort based on J indices of the extensor digitorum longus and extensor hallucis longus. Figure 5 presents the inversely proportional exponential dependencies of the Hc and Hcc parameters on the J index. The optimal areas of the force effort are highlighted with a continuous line. The range of each optimal area is about 30 kg/s. The force effort evolved, during the two-week cycles, from 10 to 35 sets of exercises. At the same time, the explosive strength changed from 500 kg/s to 170 kg/s in group I, from 550 to 220 kg/s in group II and from 600 to 280 kg/s in group III. The optimal areas of the force-velocity effort are marked with a dotted line. The range of each optimal area represents about 55 kg/s. The force-velocity effort increased, within the two-week cycles, from 15 up to 55 sets of exercises. The explosive strength had the following changes: in group I from 510 kg/s to 175 kg/s; in group II—from 555 to 220 kg/s; in group III—from 605 to 280 kg/s. The range of optimal areas of the force effort and force-velocity effort were determined taking into account the parameters dispersion values in each group of sportswomen. The area of positive correction is located above the optimal areas, which shows the increase of the training effort when exceeding the optimal area value, multiplied at the corresponding value coefficient. Below the optimal areas is located the negative correction area, which represents the decrease of the training effort to the size of the deviation for reducing the functional parameter value related to the negative area of effort.

Figure 5 Optimal areas of force effort and force-velocity effort depending on the J indices of the extensor muscles of the leg and the extensor muscles of the foot plantar area.

(A) Extensor digitorum longus and extensor hallucis longus. (B) Triceps brachii and biceps brachii. (C) Latissimus dorsi muscle Notes. OHc, Optimal areas of force effort; OHcc, Optimal areas of force-velocity effort; groups I, II, III.

Figure 5B shows the optimal areas of the force training effort dependent upon the J index of the triceps brachii and biceps brachii. The exponential dependencies of the Hc and Hcc parameters on the J index are presented in Fig. 5B. The optimal areas of force training effort are highlighted with a continuous line. The range of each area is about 20 kg/s. The force effort, during the two-week cycles, evolved from 10 to 35 sets of exercises. The muscle explosive strength changed from 120 to 45 kg/s in group I, from 140 to 65 kg/s in group II and from 160 to 85 kg in group III. The optimal areas of force-velocity effort are marked with a dotted line. The range of each area is approximately 25 kg/s. The explosive strength of muscles had the following variations: in group I from 125 to 50 kg/s; in group II—from 145 to 70 kg/s; in group III—from 165 to 90 kg/s. The optimal areas of efforts for triceps brachii and biceps brachii almost coincided, so they are shown in Fig. 5B by a single curve for each group of athletes. Above the optimal areas is situated the positive correction area, which shows the increase of the training effort. Below the optimal areas is located the negative correction area, which indicates the decrease of the training effort.

Figure 5C presents the training effort optimal areas determined by the J index of the latissimus dorsi muscle. The figure shows the exponential dependencies of the Hc and Hcc parameters on J index. The force effort areas are marked with a continuous line. The range of each area is about 30 kg/s. The explosive strength of the muscles within the two-week cycles changed as follows: from 200 to 65 kg/s in group I; from 230 to 90 kg/s in group II; from 256 to 110 kg/s in group III. A dotted line highlights the optimal areas of force-velocity effort training. The range of each zone represents approximately 30 kg/s. The explosive strength of the muscles oscillated from 205 to 70 kg/s in group I, from 235 to 95 kg/s in group II and from 260 to 110 kg/s in group III. The area of positive correction (which represents the increase of the training effort) is located above the optimal areas of the effort. The negative correction area, which means the decrease of the training effort within the two-week cycle, is situated below the optimal areas of effort.

The results of the corrections had a positive impact, leading to the considerable increase of judokas’ sports mastery in both experimental groups in terms of improved force and force-velocity capacity (Tables 2 and 3).

Table 2 Results of testing of the judokas in groups A and B (categories I and CMS) after one year of the experiment (mean ± SD).

Indices	Testing	Mean ± SD	Cohen’s d	p-value	
		Group A	Group B			
F1 (kg)	initial	88.6 ± 3.69	89.2 ± 3.60	−.25	.565	
final	95.3 ± 3.23	92.0 ± 3.32	1.01	.029*	
F2 (kg)	initial	138.4 ± 3.35	138.2 ± 3.60	.06	.904	
final	145.7 ± 3.04	142.1 ± 3.30	1.13	.003**	
F3 (kg)	initial	95.4 ± 3.56	97.2 ± 3.79	−.49	.260	
final	102.9 ± 3.30	98.9 ± 4.11	1.07	.02*	
F4 (kg)	initial	58.2 ± 3.34	57.8 ± 4.07	.11	.821	
final	63.1 ± 3.11	59.1 ± 3.36	1.24	.009**	
F5 (kg)	initial	54.8 ± 3.60	55.2 ± 3.40	−.11	.810	
final	60.7 ± 3.40	59.0 ± 3.32	.51	.242	
F-V1 (kg)	initial	183.4 ± 3.11	183.2 ± 3.19	.06	.894	
final	194.8 ± 2.64	188.2 ± 3.42	2.16	.000***	
F-V2 (sec)	initial	10.54 ± 0.72	10.68 ± 0.74	−.02	-.648	
final	10.11 ± 0.58	10.49 ± 0.71	−.59	.184	
F-V3 (sec)	initial	11.32 ± 0.68	11.26 ± 0.64	.09	.873	
final	10.22 ± 0.51	10.82 ± 0.63	−1.05	.024	
F-V4 (sec)	initial	14.40 ± 0.33	14.20 ± 0.37	.57	.200	
final	13.31 ± 0.34	14.00 ± 0.33	−2.06	.000***	
Notes.

Indices Force: F1, force of the back lumbar muscles, kg; F2, force of thigh muscles, kg; F3, force of gastrocnemius muscles, kg; F4, force of biceps, kg; F5, force of triceps, kg. Indices force –velocity: F-V1, explosive strength of lower limbs muscles, kg; F-V2, five throws over shoulder, sec; F-V3, five throws with turning and foot sweep, sec; F-V4, rope climbing 4 m, sec. F, force; F-V, force - velocity; SD, standard deviation; Cohens d effect size, d = 0.2 be considered a ‘small’ effect size, 0.5 represents a ‘medium’ effect size and 0.8 a ‘large’ effect size; t, Independent Samples Test.

* p < 0.05.

** p < 0.01.

*** p < 0.001.

Table 3 Results of testing of the judokas in groups C and D (MS category) after one year of the experiment (mean ± SD).

Indices	Testing	Group C	Group D	Cohen’s d	p-value	
F1 (kg)	initial	112.4 ± 3.04	113.2 ± 3.34	−.25	.555	
final	122.7 ± 2.97	116.3 ± 3.52	1.97	.000***	
F2 (kg)	initial	146.8 ± 4.04	147.2 ± 3.34	−.11	.821	
final	158.8 ± 2.93	151.4 ± 3.41	2.33	.000***	
F3 (kg)	initial	116.2 ± 3.31	117.4 ± 3.54	−.35	.415	
final	123.9 ± 1.51	119.9 ± 2.54	1.91	.000***	
F4 (kg)	initial	69.6 ± 4.61	68.6 ± 5.22	.20	.670	
final	74.00 ± 4.40	70.5 ± 3.62	.87	.058	
F5 (kg)	initial	64.6 ± 2.84	65.2 ± 3.34	−.19	.507	
final	68.5 ± 3.61	67.4 ± 3.56	.31	.484	
F-V1 (kg)	initial	204.4 ± 3.75	205.5 ± 3.98	−.28	.516	
final	216.0 ± 3.32	211.6 ± 4.03	1.19	.012*	
F-V2 (sec)	initial	10.20 ± 0.72	10.00 ± 1.17	.21	.661	
final	9.41 ± 0.50	9.85 ± 1.04	−.54	.344	
F-V3 (sec)	initial	11.00 ± 0.78	10.88 ± 0.92	.14	.748	
final	10.22 ± 1.08	10.56 ± 0.94	−.34	.740	
F-V4 (sec)	initial	13.40 ± 0.81	13.20 ± 0.79	.25	.566	
final	12.48 ± 0.83	13.08 ± 0.55	−.85	.060	
Notes.

Indices Force: F1, force of the back lumbar muscles, kg; F2, force of thigh muscles, kg; F3, force of gastrocnemius muscles, kg; F4, force of biceps, kg; F5, force of triceps, kg. Indices force –velocity: F-V1, explosive strength of lower limbs muscles, kg; F-V2, five throws over shoulder, sec; F-V3, five throws with turning and foot sweep, sec; F-V4, rope climbing 4 m, sec. F, force; F-V, force - velocity; SD, standard deviation; Cohens d effect size, d = 0.2 be considered a ‘small’ effect size, 0.5 represents a ‘medium’ effect size and 0.8 a ‘large’ effect size; t, Independent Samples Test.

* p < 0.05.

** p < 0.01.

*** p < 0.001.

The analysis of the data included in Table 2 shows the results of the female judokas belonging to the experimental group A (category I and CMS) at the beginning and end of the research. At the same time, the force indices changed considerably, by 52.15% (p < 0.01; p < 0.05) and the force-velocity indices by 6.22% (p < 0.05) and 6.18% (p < 0.05; F-V4, p > 0.05), which characterizes the level of physical training. The lack of scientifically substantiated corrections in the training process of the control group athletes entails the insignificant increase of the indices recorded. Thus, the comparison of the results obtained by the female judokas in the control group B (categories I and CMS), at the beginning and the end of the experiment, showed an unobservable tendency to improve the tested indices, namely by 2.83% at force (F1-5) and by 2.73% (F-V1) and by 3.85% (F-V2-4). Regarding the differences between the averages in groups A and B, one can notice significant differences in final testing at the indices of F1-4 (p < 0.05, p < 0.01) and at the indices F-V1 and F-V4 (p < 0.001). Although all the analyzed indices increased to some extent, the low efficiency of the traditional methods of judokas’ training was quite obvious from the initial stage of the experiment.

A similar situation can be found out when analyzing the testing results of the judokas in the experimental group—MS (Table 3). In this group it can be noticed, first of all, the significant increase of the force by 7.52% (p < 0.01; p < 0.05 and F4, p < 0.05) and force-velocity by 5.67% (p < 0.05) and 7.20% (p < 0.05; F-V4, p > 0.05), which indicates the level of physical training. The findings mentioned above refer also to the athletes of the control group—MS. In this group, the test results proved to be even lower: increase of force by 3.93% (p > 0.05) and force-velocity by 2.97% and 1.73% (p > 0.05), which clearly proves the stability of the indices of the physical training level in the elite sportswomen. As for the differences between the averages of the groups C and D, there are significant differences in the final testing at the indices of F1-3 (p < 0.001) and at F-V1 (p < 0.05). The insignificant increase of the physical training indices also leads to analogous tendencies regarding the criteria of the technical-tactical mastery, which also stabilized and almost did not change at the end of the research.

Discussion

The scientific innovation brought by this paper is the determination of the elite female judokas’ physical and functional reserves for the first time. The interconnection between the dynamics of the athletes’ musculoskeletal system state and the adjustment of the force and force-velocity efforts was also highlighted. The range of optimal values related to effort correction was established according to the data on the functional state of the judokas’ neuromuscular system and the medical-pedagogical examination results. The system of implementing the corrections of force and force-velocity efforts in the female judokas’ training process was approved experimentally.

Data on the change of NMS functional state of each sportswoman-subject of the research were obtained, according to the efforts of force and force-velocity, throughout the training and competition periods. Subsequently, the correlation coefficients (r) between the indices defining the NMS functional state and the adjustment of the force and force-velocity efforts were calculated by the derivation method. The correlative analysis of the tested parameters helped to determine the percentage ratio between the indices showing the NMS functional state of the judokas of category I (group I), CMS (group II) and MS (group III), depending on the total value of the force and force-velocity training effort. Therefore, the analysis of the judokas’ NMS functional state at Tmax, Fy and J indices of the main muscle groups revealed the presence of significant correlations between these parameters and the value of force and force-velocity efforts. The data reveal important differences of the determined percentage ratio depending on the athletes’ mastery and the type of the training efforts. At the same time, there is a general trend of significant increase in the value of Tmax index when the sports mastery increases. A distinctive feature of the efforts influence on the NMS functional state lies in the diminution of the Fy index value during the force effort (32% - for the athletes of category I; 30% - for the candidates to Masters of Sport; 27% - Masters of Sport) and its insignificant increase (from 28% to 30%) during the force-velocity effort. Corrections are needed when an imbalance between two groups of muscles is found out, which could impede the execution of the technique and could increase the risk of injury. In such a case, workouts with the balance board are proposed (e.g., Drid et al., 2010).

The results obtained in conformity with the dependency of Tmax index on the parameters of force effort and force-velocity effort reveal a significantly high degree of indices correlation (R = 0.548–0.854 according to the training cycle and the tested muscle). When examining the results of the elasticity indices evaluation in different muscles (Fy) related to the force-velocity effort, it was found out that the correlation coefficient value of these indices rarely has a significant nature (R = 0.509–0.852). The insignificant changes of the Fy index are practically not dependent upon the cycles and periods of the sports training. Lower values of the correlation coefficients were attested when analyzing the dependency of the explosive strength indices (J) on the force effort and force-velocity effort. In the given case, the great majority of the correlation coefficients had significant variations (R = 0.573–0.764, p < 0.05). At the same time, some variations of the coefficients recorded during the test are insignificant. The isokinetic analysis of the knee pair of flexors and extensors in the elite judo female athletes can prevent lateral asymmetry, thus reducing the risk of injury (e.g., Blach et al., 2021). The changes in the toes strength as a result of judo successive attacks and their relationship to the lactate production were also analyzed (e.g., Bonitch-Domínguez et al., 2010). There were determined the differences in hamstrings-to-quadriceps (H/Q) peak torque ratios evaluated at different angular velocities between men and women who participate in judo, handball or soccer. They suggest that sport modality and angular velocity influence the isokinetic strength profiles of men and women (e.g., Andrade et al., 2012).

The plan for each training cycle was divided into two-week periods. Force and force-velocity efforts were assigned to each training cycle, according to the number of sets of exercises. The sports classification of the subjects in the experimental and control groups was consistent with the model-characteristics created. The optimal areas of force training effort and force-velocity training effort were identified in accordance with these model-characteristics and the plan of force and force-velocity efforts. In the experimental groups, the individual correction of the force effort and force-velocity effort was performed every two weeks, based on the data resulted from the judokas’ NMS functional state testing. When examining the changes of the football players’ central nervous system state (CNS), different states (levels of fatigue) were determined during their preparatory period, depending on the training task (Bashkin & Kharlamov, 2015). Determining the adaptation areas individually built through both positive and negative correction allows the coach to adjust the training tasks for different periods of training (Kharlanov, 2015; Kharlanov, 2021). The integrative neuromuscular training can be used to enhance the injury endurance and to improve the sports and motor performance skills in young people (e.g., Fort-Vanmeerhaeghe et al., 2016).

The correlations between the modification of the functional parameters and the training effort within the two-week cycles helped to identify the model-characteristics of the optimal sizes in terms of force and force-velocity training efforts. The model-characteristics were represented by optimal areas, determined using the average values of the functional parameters Tmax, Fy and J in each training cycle, for each group of subjects separately (Figs. 3, 4 and 5). There are differences in terms of maximum and explosive strength and the decrease ratio of the peak force between the dominant hand and the non-dominant one. The effects of the training with various types of mechanical loads on the muscle force (F), velocity (V) and power (P) were monitored (Detanico et al., 2012; Djuric et al., 2016).

The number of positive and negative force corrections for each athlete within a monthly cycle change from 0 to 3 (according to the number of functional parameters—Tmax, Fy and J). A minimum number of force effort corrections were done along the first two training cycles, fact entailed by athletes’ NMS adaptation processes. A maximum number of positive and negative corrections were done in the sixth training cycle (the highest volume of force effort—35 sets of basic exercises—is achieved within this cycle). At the end of each training cycle, the total number of corrections decreased twice compared to the maximal value. At the same time, the force effort consisted of 16 sets of exercises. The total number of negative corrections in the experimental groups was 79 and the positive ones—59 (group A); as for the group C, a number of 65 negative corrections of the force effort and 89 positive corrections were done. In some cases, the negative corrections of the training efforts in both experimental groups were replaced by recovery activities (rest, sauna, massage), and in other groups the force effort was diminished by 10–50% (group A) and by 10–45% (group C). When making positive corrections, the physical effort was intensified by 10–50%. The total volume of force efforts during this period, after making the negative and positive corrections, did not change significantly (p >0.05). Within the force-velocity effort, 68 negative corrections and 92 positive ones were made in group A, while in group C there were 81 positive corrections and 78 negative ones throughout the entire period. Sometimes, the training sessions were replaced by activities of recovery (rest, massage, sauna). Within the correction, the force-velocity effort was reduced by 10–45% and increased by 10–50%. The total volume of force-velocity efforts, after making the positive and negative corrections, did not change significantly (p > 0.05). There were examined the differences in force-velocity characteristics and the maximal power and rise height of the body’s centre of mass, measured in the counter-movement jump (CMJ) and the spike jump (SPJ), in the case of the judokas, boxers and taekwondo athletes (e.g., Buśko, 2016). The exploration of the force-velocity (F-V) relationship model in the leg muscles and the evaluation of the concomitant validity and reliability of the obtained parameters were also performed. The changes associated with the load in muscle work and the output power (e.g., Feeney et al., 2016) and the relationship between the vertical mechanical profiles (jumps) and the horizontal ones (sprinting) in terms of force-velocity-power (FVP) were studied for a large range of sports (e.g., Jiménez-Reyes et al., 2018).

Also, the force and force-velocity indices changed considerably, as well as the test results that characterize the level of physical training. The lack of scientifically substantiated corrections in the training process of the athletes from the control group entails the insignificant increase of the recorded indices. Thus, the comparison of the test results in the judokas from the control group B (categories I and CMS), at the beginning and at the end of the experiment, revealed an unnoticeable tendency to improve the tested indices (Table 2). High levels of muscle power are required to optimize the training programs of the judokas, accompanied by a moderate use of the aerobic metabolic pathway, which is consistent with the features of the judo (e.g., Sbriccoli et al., 2007). The isokinetic tests of the male judokas’ knees demonstrated that the asymmetries are better detected by eccentric testing (e.g., Šimenko, Karpljuk & Hadžić, 2022).

The considerable improvement in the physical training indices and in the capacities of force and force-velocity leads mainly to a higher standard of the competitive combats and to better results of the technical actions.

Limitations of the study

This is an experimental type research, which is composed of two parts: the diagnosis of the NMS functional state in the elite female judokas of different classification categories at the beginning of the research; the experimental part carried out in two stages, with two experimental groups and two control ones. The individual results and their presentation per group averages of the studied indices is a limit of the research. Due to the small number of elite judokas aged 20–25 years at each classification category and weight category, the results obtained have scientific significance and relevance only at research level and cannot be generalized. However, they have a great practical applicability for judo and other combat sports as well, where the effort of force and force-velocity is a determining factor in achieving sports performance. Because there is no database with the values of the NMS indices in the elite female judokas, the research results were analyzed only by comparison between classification categories and control groups, without reference to the elite athletes’ values, especially the elite female judokas.

Conclusions

The functional state of athletes’ neuromuscular system and their level of special training must be determined through the efficient implementation of advanced technologies and mobile computing resources. Thus, the data collection and processing, the objective and fast displaying of the results, the organization and direct development of the specialized evaluations in training conditions are speeded up.

The implementation of the individual correction plan in the instruction and training plan for the elite female judokas must be based on the exact determination of the force efforts and force-velocity efforts throughout the training and competitive periods.

The functional state of the female judokas’ neuromuscular system, determined on the basis of the temporal indices of the maximal muscular contraction, the explosive strength and the muscles elasticity is subjected to an effervescent dynamic under the influence of the force efforts and force-velocity efforts.

Supplemental Information

Supplemental Information 1 Raw data

The percentage ratio of the value of the neuromuscular system indices of the female judokas with different levels of qualification, depending on the force and force-velocity effort. The number of corrections of the force effort and force-velocity effort applied in the training of the female judokas of the experimental group are presented. Also presented are Optimal areas of force effort and force-velocity effort based on the Tmax indices of the tested muscles, Optimal areas of force effort training (A) and force-velocity (B) depending on the Fy indices of the tested muscles and Optimal areas of force effort and force-velocity effort depending on the J indices of the extensor muscles of the leg and the extensor muscles of the foot plantar area.

Click here for additional data file.

This article is part of the research project no.15.817.06.26F on the: “Scientific-methodical support of the training process of the national teams of the Republic of Moldova for the Olympic Games, European and World Championships” –State University of Physical Education and Sport, Chisinau, Republic of Moldova. We also express our gratitude to the coaches of the women’s judo national team of the Republic of Moldova and to the athletes who participated in this research.

Additional Information and Declarations

Competing Interests

Author Contributions

Human Ethics

Data Availability

The authors declare there are no competing interests.

Veaceslav Manolachi conceived and designed the experiments, performed the experiments, analyzed the data, prepared figures and/or tables, authored or reviewed drafts of the paper, and approved the final draft.

Vladimir Potop conceived and designed the experiments, performed the experiments, analyzed the data, prepared figures and/or tables, authored or reviewed drafts of the paper, and approved the final draft.

Victor Manolachi conceived and designed the experiments, performed the experiments, analyzed the data, prepared figures and/or tables, authored or reviewed drafts of the paper, and approved the final draft.

Irina Delipovici performed the experiments, analyzed the data, authored or reviewed drafts of the paper, and approved the final draft.

Cristian Ştefan Liuşnea analyzed the data, prepared figures and/or tables, and approved the final draft.

The following information was supplied relating to ethical approvals (i.e., approving body and any reference numbers):

The Ethics Commission of the State University of Physical Education and Sports of the Republic of Moldova granted approval to carry out this study (ec_usefs4-11-2021).

The following information was supplied regarding data availability:

The raw measurements are available in the Supplementary File.

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
