# Peer review of "Determining the optimal areas of effort in terms of force and force-velocity based on the functional state of the neuromuscular system in the training of elite female judokas"

_PeerJ, doi:10.7717/peerj.13468_

## Round 0.1 · original submission · Major Revisions

Thank you for submitting the manuscript to PeerJ. It has been reviewed by experts in the field and we request that you make major revisions before it is processed further.

We look forward to hearing from you soon.

Best wishes,

Badicu Georgian, Ph.D

·

Basic reporting

Thanks for sending through the manuscript entitled " Determining the optimal areas of effort in terms of force and force-velocity based on the functional state of the neuromuscular system in the training of elite female judokas".
I went through the manuscript and I found that overall it is suitable for publishing in the Peer J, but I have some major methodological concerns and other issues that the authors need to address before I can accept the manuscript for publication.
The problem is the lack of proper familiarity with the English language and recommend that the entire article should be corrected by someone who is a proficient editor.

Abstract
• The info should be better presented as to raise the interest of the readers. The more data the authors give, the more difficult the abstract is to read and the interest to be risen.

Introduction
• The introduction is incomplete and the issue is not well clarified and also the need for research is not clear. The rationale for examining this topic should be mentioned more clearly in this section. The article innovation should be also presented in the Introduction. Describe what the research gap of the paper is and what is new. Please describe the links between the research gap and the goal of the article. At least the reader should be given a background on how each of the identified factors are important and informed on the novelty of this study.

• The second sentence of the first paragraph contains too many ”of”, which make it difficult to read and understand. This is the case for many parts of the article (for example, starting with lines 103-104… ending with 567-569) as well as other mistakes (line 114 …”the more important”?)

• Lines 136-138 are not enough to make your study of sound scientifically importance and it is the exact title of this article also. Please provide a subsection here with the title RESEARCH QUESTIONS, AIMS OF THE STUDY or RESEARCH HYPOTHESES, which would be more convincing about the importance of the study.

Experimental design

Participants and methods

• It is not clear how many female judokas were involved in the study, were there 21 or 44? Were those from the three groups (I, II, III) the same with those in EG and CG? First there are mentioned 44 cycles and only 12 training cycles and 12 competitive ones.

• Authors should consider trying to make explanations less boring and shorter here. A table or illustration of some kind would be more attractive and easier to follow.

Validity of the findings

The Results section should be reorganised as to follow each hypothesis/research question or objective. Authors need to write key findings focusing on each one of these after being stated.

Table 3 contains groups A and B (as it is in Table 2) but the explanations are about groups C and D.

The Discussion section is centred on the correctional system. Is it of great importance? If so, this should be the central idea of the article and the title as the discussion focuses on numbers of corrections along the cycles.

It would be better to have seen more use of terms like 'originality' and 'significance'. Identify what is new in this study that may benefit readers or how it may advance existing knowledge or create new knowledge on this subject. There should be a clear conclusion on why the research findings are significant for this topic and how could be used to help athletes in this situation.

Only when reading lines 576-578 in the Conclusion section readers can understand the intention to create an optimal model of F and F-V training efforts derived from NMS.
Research limitations and existing problems are not presented.

Additional comments

Thank you for the opportunity to read this work. This is an interesting topic that can be considered by readers. Nevertheless, there are some concerns with the present manuscript that would need to be addressed for the paper to be able to achieve its potential.
Though the paper seems properly organised, it is not and easy to follow. Some issues are not clearly defined or presented (e.g., hypotheses/research questions, procedure), which makes it difficult to understand the various parts of the manuscript.
The problem is the lack of proper familiarity with the English language and recommend that the entire article should be corrected by someone who is a proficient editor.
The introduction is quite incomplete and the issue is not well clarified and also the need for research is not clear.
In the method and measurement section, the work process is very incomprehensible and incomplete and makes it difficult to understand.
The conclusion section is very weak and the reasons presented are not well compared with previous literature and other articles.

·

Basic reporting

Although the manuscript is mainly nicely written in English, I suggest rechecking the grammar.
Literature references and sufficient field background/context are sufficiently provided.
Tables 2 and 3 have to be modified. I suggest removing the variables as “ES r” ,“t”,”confidence interval of the difference”. And p value should be written only as “p value”, “Sig. two-tailed” should be deleted.
The quality of figures 3 and 4 should be improved.
Figure 3. “Inner femoris muscle” – Using Latin names is suggested. Check elsewhere in the manuscript.
Raw data is partly shared.

Experimental design

The research is original and follows the aims and scope of the journal.
The entire methods section needs to be rechecked and reviewed. Data regarding the number of participants and the duration of the study seem quite confusing.

Validity of the findings

The conclusion should be shortened by mentioning only the most important findings of the current study.

Additional comments

By reading the abstract, it is unclear how many participants are included in the study (21 or 44). Moreover, the authors in methods have mentioned 4 groups A, C, B, and D, of which A and C are labeled as experimental and B and D as control groups. Why do the groups have to be in this order? Is there a particular reason? Then, in the results, the authors stated 1st group results, which is unclear since the mentioned groups were A, C, B, and D. Abstract needs to be modified.
No strengths and limitations are presented in this study.
Ln33 – Why are insignificant results presented within the abstract?
Ln44 – “The F increased considerably” – Only statistically significant results should be presented in the abstract.
Ln64 – “SJFT” – Is this test used in this study as a method? There is no sense in describing some tests extensively if it is not related to the aim of the research.
Ln82 – “(e.g. Busko and Nowak, 2008) – Why “eg.”?
Ln83 – “in the judo” – “the” is not necessary. It can be deleted.
Ln113-115 – “All this is all” – sounds incorrect. Please recheck the meaning of the sentence and modify it. You are stating that women have become full participants in various competitions of the highest level. The question is, is this something that happened recently?
Ln115 – “results obtained by them in a number of sports almost equal those of men.” - What kind of the results? This statement is quite confusing.
Ln120,121 – This sentence should be rephrased.
Ln126-130 – Why do you think these studies are important to be mentioned in the introduction?
Ln131 – “local scientific literature” – What do you mean by local scientific literature?! The literature is either scientific or not. Please clarify the term local. “number of works dedicated to one or another aspect of female judokas’ sports training.” – This should be rephrased.
Ln143-145 – cg.I, CMS, MS – these categories are not sufficiently explained. As far as I know, judokas are categorized according to belts. What are the criteria for categorization? Please provide an explanation.
Ln146 –“23 – 32 years old” – No demographic characteristics of judokas presented. I suggest presenting the mean and standard deviation. Additionally, there is no information regarding their weight categories available.
Ln169 – “gastrocnemius muscles, rectus femoris muscle, thigh biceps, arms biceps and triceps.” – I suggest using Latin names of muscle. Please check everywhere throughout the manuscript.
Ln167,177, 188 – In the methods, the three tests were listed: Tmax, Fy, and J. However, the protocols are later not explained in the same order. I suggest correcting it.
Ln213 – “shoulder throws” – Are there similar throwing techniques in judo? Why have authors chosen wrestling shoulder throws?
Ln237 – experimental design – This section should be written in a more understandable way. Inserting schemes/ photos of the protocol could help readers clearly understand the study design.
Ln285. Please rewrite as: Independent-Samples T-test.
Ln383,384 – “from de la 510 kg/s” – What is the meaning of “de la”?
Ln442,443 – “…during the first year of the experiment.” – Again, it has to be clarified what the total duration of the experiment was?
Ln490 – “vibrations” – Please check if this term is correct.
Ln491 – “izokinetic” - isokinetic.
Ln545,546 – Please explain why this is not defined within the methods section?
Ln576,577,578 – Needs to be rephrased.

---

## Round 0.2 · Minor Revisions

I am writing to inform you that your manuscript it requires a number of Minor Revisions.

We are waiting the requested changes.

Kind regards,

Badicu Georgian, Ph.D
Academic Editor, PeerJ

·

Basic reporting

Thank you for providing this comprehensive and hard work.
The authors have presented an improved version of the manuscript.

The introduction provides a proper background of the topic. The manuscript is well-structured, and it is easy to follow the sections.
The quality of the images is good enough, but I don’t know if the reviewing version has lower resolution than the final version. If not, images should have better resolution in its final size.
It seems that the English is technically correct, though some statements are too long.
Lines 302 and 372 contain some green letters.

Experimental design

The experimental design meets the scope of the journal, and the hypothesis is well-defined and its relevant to the community.
Methods are described detailed enough.

Validity of the findings

The results and the conclusions are quite interesting and well-discussed. All data are provided.

The authors have adequately addressed all my comments. I have no further suggestions.

·

Basic reporting

OK

Experimental design

Ok

Validity of the findings

OK

Additional comments

Ln177 – rephrase please. For example: “Women’s training should be planned differently…”
Conclusion – should be rewritten in a shorter form. Present only the most important information here (example: no need for explaining the whole study design in this section=.
Ln838 – “achievd” – achieved.
Ln839 – limitations – please consider placing the study limitations right after the discussion.

---

## Round 0.3 · accepted · Accept

Thank you for submitting the manuscript to PeerJ. Great improvements were performed in the manuscript. Currently, the article is acceptable for publication.

We look forward to hearing from you soon.

Best wishes,

Badicu Georgian, Ph.D
BMC Public Health